# A cross-country and country-specific modelling of stock market performance, bank development and global equity index in emerging market economies: A case of BRICS countries

Ebere Ume Kalu[1]*, Augustine C. Arize[2], Okoro E. U. Okoro[1], Florence Ifeoma Onaga[1], Felix Chukwubuzo Alio[1]

1 Department of Banking and Finance, Faculty of Business Administration, University of Nigeria, Nsukka, Enugu State, Nigeria, 2 Economics and Finance Department, College of Business, The A&M University–Commerce, Commerce, Texas, United States of America

* ebere.kalu@unn.edu.ng

**Data Availability Statement:** The Data used were drawn from World Development Indicators and are

## Abstract

This study investigated in cross-country and panel form the interactions of bank development, stock market development and global equity index, focusing on the BRICS countries covering the period 1990 to 2018. We found a bidirectional causation between bank development (CPSGDP) and stock market performance as proxied by the depth of the markets (MCAPGDP) in the BRICS countries. Cointegration was also found using the panel cointegration framework and the bounds test for the ARDL estimators. This largely proves that a long-run relationship of both direct and reverse nature exists between bank development and stock market performance. For the bank development and market performance models respectively, all the error-correction terms were found to be negatively significant, indicating that they both share dynamic profile and adjust appreciably to deviations from equilibrium between the short run and the long run. The global equity index showed that stock market development interacts more with the global financial environment than bank development in the BRICS countries. Our findings support the complementarity and coevolution hypothesis in the stock market and bank development nexus.

## 1. Introduction

Financial development is characterized by improvements in the financial system in terms of monitoring firms, effective corporate governance, management of risk, mobilization and pooling of savings, provision of information about investments and allocating capital, trading, diversification [1, 2]. Following [3] financial development represents a mixture of financial depth and liquidity of the financial markets, the capability of firms and households to access financial services and the ability of financial institutions to provide services at affordable cost

available at https://databank.worldbank.org/reports.aspx?source=world-development-indicators.

**Funding:** The author(s) received no specific funding

**Competing interests:** No competing interest(s)

and sustain earnings. Bank development and market development have been the two strands of causal factors upon which financial development rests. [4] connects institutional factors, macroeconomic policies and environmental factors as potential determinants of financial development. Strong legal and regulatory institutions with improved practices are also seen as key to financial development [5].

The complementarity and the substitutability arguments of stock market development and bank development have been apprised by existing literature with inconclusive evidence. Against this background, this study empirically investigates the relationship between bank development and stock market performance in emerging markets, with emphasis on BRICS countries (Brazil, Russia, India, China, and South Africa). In this study, bank development is measured by credit facilities to the private sector scaled by the Gross Domestic Product (CPSGDP), and stock market development is measured by the market size and liquidity indicators and is proxied by market capitalization scaled by the gross domestic product (MCAPGDP) (see [6]). Stock markets and banks alike have over the years been seen as key drivers of not just the growth and development of economies but also of financial systems. This makes them command strong policy considerations in emerging and developed economies alike. Recent arguments have focused on the degree and direction of influence of the markets and banks as drivers of economic growth and financial development.

Another line of argument has become the interdependence of banks and stock markets in this drive for financial system growth and development. This is the predisposition of this study, but more significantly from the geographical standpoint of a collection of strong and emerging economies in the like of the BRICS countries. Also of investigative interest is the interaction of stock market and bank development in the BRICS countries with the global equity market. It is evident that banks and stock markets have gone beyond borders and that interconnectedness with other markets and financial systems can create some transmission and spatial effects on local financial system performances. Global equity performance indicators such as the S&P Global Equity Index have been used by prior studies to measure the global equity market impact on domestic stock market performance (See [7, 8]).For the purpose of this study, we used the S&P Global Index to investigate BRICS countries and global equity market interface given that it is a combination of three key indices, namely, The S&P Frontier Broad Market Index, The S&P Global Broad Market Index and the S&P/IFCI. Also, it includes the most liquid and investable stocks in developed, frontier and emerging markets while representing all regional, style, sector and size-based equity markets.

In the light of the above, our enquiry is driven by some key considerations:

First, BRICS countries are major recipients of global investment flows and are among the main global consumers of commodities. Changes in the global economic factors are known to be channels through which fluctuations in the world economic and financial conditions (such as the global financial crisis of 2007/2008) are transmitted to the BRICS stock markets. Knowing how this interfaces with the depth and development of the financial system is of investigative interest. Moreover, international investors are especially interested in the BRICS stock markets' co-movements with these global factors because investment, speculation and risk diversification opportunities may arise. It is obvious that financial development has come to represent a strong consideration in choosing investment destinations. Therefore, the extent to which developments in the financial system are shaped by the depth, liquidity and activities of the stock market has come to be of interest to regulators and market operators.

Second, this study addresses three-pronged questions. First, can dependence be established between bank development and the performance indicators of the stock market of the BRICS countries by using panel framework? Second, to what extent can co-movement be established between bank development and stock market performance in BRICS countries in the face of

global equity position indicators? Finally, we address the bank development, stock market development and inflation trilemma in BRICS countries. Understanding how bank development in BRICS countries is affected by domestic stock market performance and global equity stock markets while taking into account inflation, is of critical importance to policy makers and market participants.

It is common knowledge that BRIC economies came into existence in 2001, with the coining of the term BRIC to refer to the fast-emerging national economies of Brazil, Russia, India and China. With the inclusion of South Africa in 2010, the term became BRICS. Characteristically, BRICS members are all fast developing and rapidly growing with the exception of Russia, which has been an economic superpower before the uncoupling of the Union of Soviet Socialist Republic (USSR) in 1990 [9]. By way of economic influence, these economies account for close to 40% (3 billion) of the world population in 2015, with a combined nominal GDP of over US$ 16 trillion [10] and an estimated US$ 4 trillion in combined foreign exchange reserves. All BRICS nations are G-20 countries and have a combined GDP, which is approximately 20% of the world GDP. At the same time these economies have diverse cultures, different geographical and political circumstances and different rates of growth, which can come in their way of forming a strategic long-term alliance [9]. Given that economic activities are sustained and growth achieved through vibrant financial systems predominantly led by the banks and stock markets, investigating the stock market, bank development and global equity interaction in BRICS countries is of research, social and policy significance. This study is positioned to fill a gap in the literature since previous studies on stock markets performance and bank development nexus of the BRICS nations might not have captured the most recent data or introduced the global equity market interactivity indicator as is the case in this study.

In terms of individual characteristics, it is evident that the BRICS countries are of different economic and financial structures. Table 1 presents the economic and financial structures of the BRICS countries.

The five-year average bank development indicator presents China as the best developed though characterized by high public sector dominance. This is followed by South Africa with a structured and private-sector-driven open economy. India has the least bank development indicator which undoubtedly is caused by public sector dominance, low bank penetration, high cost of intermediation and difficulty in accessing credit. A look at the depth of the stock market as indicated by the five-year average of stock market capitalization, shows China as the

**Table 1. Relevant economic and financial characteristics of BRICS.**

| Countries | Economic Structure | Financial System | (2015–2019) 5-year Average of Bank Development Indicator | (2015–2019) 5-year Average of Stock Market Capitalization ($) |
|---|---|---|---|---|
| Brazil | Liberalized and market-driven | Private sector driven, growing capital market; significant inflow of foreign capital | 62.69 | 862,400,000,000 |
| Russia | Dominant Government Control with increasing drive towards market openness. | Excessive Government ownership of banks with over 50% of state-owned banks, account and banking industry assets. Gradually evolving stock market. | 52.87 | 442,800,000,000 |
| India | Dominant government control, Institutional weaknesses and constrained market development | Public sector dominated banking industry; low bank penetration; high cost of intermediation; restricted access to credit. | 49.99 | 1,936,000,000,000 |
| China | Largely Government controlled economy | Public sector dominated banking industry; with elements of directed credit. | 157.24 | 7,812,000,000,000 |
| South Africa | Market driven, Structured and Open Economy | Well-developed financial sector; Private sector dominated financial system | 115.52 | 968,400,000,000 |

Source: [11, 12].

leading stock market in the BRICS country with India following, South Africa is the third with Russia having the lowest stock market capitalization. Given the obvious disparity in the eco-financial characteristics of these countries, a study such as this unveils the interaction of bank development, stock market development and global equity index in a manner that the empirical evidence will trigger country-specific policy retooling for the imperatives of targeted bank and stock market development. In addition it will create a consciousness on the part of financial market players in the countries in the area of exploiting the opportunities that global financial interconnectedness offers. This is the reason for the inclusion of the global equity index as one of our variables in this study. Countries like India, Russia and China may be driven by the results posted by this study to consider greater openness in this era of global financialisation. Private sector and liberalized economies like Brazil and South Africa can be encouraged in the direction of greater financial deepening and inclusiveness for economic growth and development purposes.

Our analytic framework is novel and markedly different as it is dictated by some key considerations: First, consideration is given to some core preliminary statistical properties of the series by looking at descriptive statistics, the co-movement and linear association of the series and the other properties of the panel data. These tests are important first steps in specifying a model, given ultimately that there is no "true" model and that they help us to avoid specification bias [13]. Second, a panel unit root test is carried out to examine the stationarity properties of the series under study. The use of panel stationarity test is relatively recent but has been made necessary by the possible identification of time series properties in panel data (See [14–17]). For this purpose, the [14] panel unit root test is used because of its superiority in terms of power and its assumption of cross section independence [18].

Third, the study comparatively adopts the fixed effect model and random effect models for the purpose of panel data analyses. The selection of the better suited model from the two is done following the Hausman test as presented by [19]; the Hausman test represents a distance measure between fixed effect and random effect model with an $H_o$ that the random effects are better, efficient and consistent and an $H_1$ that the fixed effects are better, more efficient and consistent. Rejecting the null hypothesis implies a preference of the fixed effect model over the random effect model.

Fourth, it is a common practice to adopt panel models in cross-country studies such as this with the assumption of cross-sectional homogeneity. Such practices, though considered convenient with the attendant advantages, have been criticized because they tend to ignore cross-sectional nuances and hence are exposed to aggregation bias. Following [18], there are other problems such as measurement error distortion, selectivity problems, short-time series dimensions and others. To address the panel analyses challenges, this study in addition to the panel models used the autoregressive distributed lag (ARDL) model suggested by [20] to investigate the countries on specific basis. Therefore, this study exploits the benefits of both panel and country-specific analyses while addressing the shortcomings of each of these techniques through the combination.

Aside from the introductory elements as presented above, the rest of the paper is structured thus: section two documents a review of literature, section three shows the adopted empirical techniques, while section four contains results and findings and is followed by section five showing summary, conclusions, and policy implications of the study.

## 2. Review of literature

Sequel to the position established by prior authors and the need to fill the discovered gaps in literature, this study follows the understated null hypotheses:

$H_{o1}$: *Bank development does not drive stock market development*
$H_{o2}$: *Stock Market development does drive bank development*
$H_{o3}$: *Global equity index has no significant impact on bank and stock market development.*

For close to two decades, there have been debates regarding the superiority, complementarity and substitutability of bank-based over market-based financial systems as catalysts for economic growth and development [2]. Some authors have advocated for bank-based financial systems over market–based systems, and others have stood for market-based systems.

A bank-based system has been supported on the grounds that it can induce longer-term investment in the real sector given that banks can enhance domestic investment [21]. Such proponents have argued against the market-based system on the grounds that the market is exposed to the sensitivity of price changes and other macroeconomic influencers, which affects long-run stability [22–24] holds that bank-based systems can allow for due government control and can drive monetary policies, industrial policies and other issues that the market cannot regulate. On the other hand, supporters of the market-based system posit that banks are exposed to inefficient capital allocation, an improper nexus between banks and firms, higher debt-ratio, moral hazard and the likelihood of implicit government bailout and vulnerability to financial crisis [25, 26].

It is obvious that none of the two systems is absolute in terms of servicing the growth needs of any financial system. There is a shift of the current argument from superiority to complementarity and substitutability of the two systems. [6, 27] argue that banks and stock markets play the role of financial intermediation in any economy and that they can be seen as either substitutes or complements without the superiority argument.

[28] introduced the argument of the developmental stage of the financial system, positing that more developed financial systems would go in the direction of larger and more liquid stock markets, while the less-developed ones would favour more of a largely bank-based structure. [29], in support of the above, held that the relationship between bank-based and market-based systems is non-monotonic, holding that at the early stages of development the banking sector serves as a complement to the stock market while at a later stage the systems diverge and become competitive. This argument corroborates [30] three-dimensional theoretical basis for interactive relations between banks and stock markets, which are competitive, complementary and co-evolving interactions.

[31] present a different line of argument not centered on the superiority of the financial structure model but on the ability of each model to reduce financing cost for economic activities. [32] assert that better-developed financial systems facilitate economic growth in the long-run. Along the same line of argument, authors like [30, 33–35] conclude that, after all, no economy operates a 'pure' model and that the strength in one aspect of the financial system (say, banks) should reflect on the other (say, stock market).

In this area of investigation, some studies have focused on BRICS countries following the above lines of argument. [36] measured financial development spillover in BRICS economies using three measures of financial development, viz., equity market, money supply and market capitalization, and used the global vector autoregressive (GVAR) framework in analyzing the quarterly data covering the period, 1989Q1 and 2012Q4. The study finds that equity market and money supply do not impact the economic growth of each BRICS, whereas market capitalization significantly affects economic growth. This finding tends to support the market-based argument in financial sector development.

Introducing a new set of estimators into the discussion, [7] examined the structure of dependence shared between the stock markets of BRICS using S&P 500 stock returns, the WTI crude price, the gold price, the U.S. policy uncertainty index and the VIX index, finding that the global stock and commodity markets exert a strong impact on the BRICS stock markets, while the U.S. economic policy uncertainty has a non-significant effect on the BRICS stock

markets. This shows that exposure to the international financial system can shape the interactions between stock market and bank development given the globalized nature of the banking and stock market subsystems of financial development. Empirical evidence in support of the above dimension of the debate on the stock market and bank development abounds in such studies as [8, 37–39]; and even [40], which revealed that the Russian equity market shares strong relationship with the global stock markets.

Much as a study with global interaction perspective is necessary, [41] argue that pooling countries can affect the disclosure of country-specific nuances when looking at such groups as BRICS countries. Countries like China are highly regulated with a plethora of reforms. This may affect market and bank functioning, contrary to what obtains in other financial systems [42]. This view is corroborated by [43], who opined that financial systems should be distinguished in terms of functional strategies because such are tied to regulatory frameworks prevalent in the environment. Much as attempts are made to study countries distinctly, it is also important to reckon on the global interconnectedness of financial systems and reflect the same in relative measurements and studies.

Conclusively, our study unlike prior studies, fills a gap in the literature by not only being focused on the BRICS countries but also by following a different line of investigation from prior studies. Specifically, this study brings together bank development, stock market performance and their interaction with the global equity environment in both a country-specific and panel setting while controlling for inflation.

## 3. Methodology

### 3.1 Data and methods

The variables for this study are in their natural log form except INFR, and are drawn with attention to stock market performance, bank development and global stock index interaction, which are the focus of this study. The datasets are cross-country/panel in nature, focusing on the BRICS countries and covering the period 1990 to 2018. Credit to the Private Sector as a ratio of Gross Domestic Product (CPSGDP) We follow authors such as [1, 44–46] in using CPSGDP as a proxy for bank development including three policy/working papers by the IMF on: (i) Remittances, Financial Development and Growth by [47] (ii) Financial Development and Economic Growth by [48] and, iii. Introducing a New Broad-based Index of Financial Development by [3]; and Market Capitalization as a ratio of Gross Domestic Product (MCAPGDP) as proxy for stock market depth. The explanatory variables include Broad Money Supply as a ratio of Gross Domestic Product (M2GDP); Stock traded as a ratio of the Gross Domestic Product (STRDEDGDP); while controlling for inflation (INFR), the interaction of the BRICS countries' bank and stock market development with the global financial market is evaluated by the introduction of the S&P Global Equity Index (S&PGLOBALEQUITY) as one of the regressors. The direct and reverse causation of bank development and stock market performance is shown in this study by making it dual-modelled as shown below:

$$CPSGDP_{it} = f_0 + f_1 M2GDP_{it} + f_2 MCAPGDP_{it} + f_3 INFR_{it} + f_4 STRDEDGDP_{it} \\ + f_5 S\&PINDEX_{it} + v_{it} \tag{1}$$

$$MCAPGDP_{it} = f_0 + f_1 M2GDP_{it} + f_2 CPSGDP_{it} + f_3 INFR_{it} + f_4 STRDEDGDP_{it} \\ + f_5 S\&PINDEX_{it} + v_{it} \tag{2}$$

All the variables are as defined above and $f_0$–$f_4$ are coefficients of the estimators with $v_{it}$ given as the residual or error term.

## 3.2 The estimation approach

First, we carried out descriptive statistics on both the panel and the country-specific data sets. Aggregative tendencies are shown using such averages as the mean and median while dispersion is shown with normal standard deviation and relative standard deviation (coefficient of variation).

Second, we carried out sets of unit root test—panel unit root tests and country-specific unit root tests to expose the stationarity properties of the datasets using such tests as [14] for the panel and ADF structural break consistent test for the individual country's test.

The models for the panel unit root tests follow the form specified below:

$$\Delta Y_{it} = \alpha_i + \rho Y_{it-1} + \sum_{k=1}^{n} \emptyset_k \Delta Y_{it-k} + \delta_i t + \theta_t + u_{it} \tag{3}$$

This is with an $H_0$: $\rho = 0$ *and* $H_A$: $\rho < 0$.

The LLC test is considered suitable given that it is both time specific and entity specific and also allows for separate deterministic trend with appropriate lag structure to mop up autocorrelation [49].

For the country-specific unit root test the augmented Dickey Fuller sequential procedure for unit-root test that uses the whole sample is carried out using the following regression model with a design of selecting the break date endogenously.

$$\Delta y_t = P y_{t-1} + \mu + \alpha \vartheta_t(t_{used}) + \lambda t + \sum_{i=1}^{p} \alpha_i \Delta y_{t-i} + \mu_t \tag{4}$$

Where $(t_{used})$ = Tb/T, which is the trimmed sample.

$\vartheta_t(t_{used})$ allows for the break, which can either be in the level where it is equal to 1 when t $> t_{used}$ and 0 if otherwise. It can also break in the deterministic trend where $\vartheta_t(t_{used}) = t - t_{used}$ if $t > t_{used}$ *and 0* if otherwise.

For the purposes of carrying out this structural break consistent unit root test, firstly, the standard Dickey-Fueller test is estimated. Secondly, the minimum DF statistic $t_{DF}^{min}$ is obtained, and the maximum DF statistic $t_{DF}^{max}$ were obtained and the difference between the two (max and min) taken finally. The break date selection processes and trimming are performed to remove endpoint values from being considered as break dates.

The inducement of the structural break is evaluated whether it is gradual (innovation outlier) or rapid (additive outlier); this follows the form presented below:

Additive Outlier Model:

$$\tilde{y}_t = \sum_{j=0}^{k} W_t\, D(Tb)_{t-1} + \alpha \tilde{y}_{t-1} + \sum cj \Delta y_{t-j} + \varepsilon_t \tag{5}$$

Where: $\tilde{y}_t$ = a detrended series of Y and $Y_t = \gamma + \partial D \mu_t + \tilde{y}_t$
Innovation Outlier Model:

$$Y_t = \gamma + \partial D \mu_t + \theta D(Tb)_t + \alpha \tilde{y}_{t-1} + \sum_{j=1}^{k} cj \Delta y_{t-j} + \varepsilon_t \tag{6}$$

Third, the study comparatively adopts the fixed effect model of the panel data analyses method and random effect model. The fixed effect model according to [49] follows the form presented below:

$$Y_{it} = \alpha + \beta X_{it} + \lambda_{it} + \upsilon_{it} \tag{7}$$

$\lambda_i$ is a time-varying intercept that captures all the variables that may affect $Y_{it}$ which are time-variant and cross-sectionally constant. Substituting our variables under study into the

fixed effect model framework will appear thus:

$$LCPSGDP_{it} = f_0 + f_1 M2GDP_{it} + f_2 MCAPGDP_{it} + f_3 INFR_{it}$$
$$+ f_4 STRDEDGDP_{it} + f_5 S\&PINDEX_{it} + \lambda_{it} + v_{it}$$
(8)

$$MCAPGDP_{it} = f_0 + f_1 M2GDP_{it} + f_2 CPSGDP_{it} + f_3 INFR_{it}$$
$$+ f_4 STRDEDGDP_{it} + f_5 S\&PINDEX_{it} + \lambda_{it} + v_{it}$$
(9)

$\lambda_i$ is a time-varying intercept, $v_{it}$ is the error term.

The random effect model according to [49] as presented below:

$$Y_{it} = \alpha + \beta X_{it} + \omega_{it}, \omega_{it} = \varepsilon_{it} + \mu_{it}$$
(10)

where:

$\varepsilon_{it}$ measures the random deviation from the global or common intercept term $\alpha$, subscript "***it***" represents the combination of individuality and time and $\mu_{it}$ is the error term.

Substituting our variables into the fixed effect model will appear thus:

$$CPSGDP_{it} = f_0 + f_1 M2GDP_{it} + f_2 MCAPGDP_{it} + f_3 INFR_{it} + f_4 STRDEDGDP_{it}$$
$$+ f_5 S\&PINDEX_{it} + (\mu i + \varepsilon it)$$
(11)

$$MCAPGDP_{it} = f_0 + f_1 M2GDP_{it} + f_2 CPSGDP_{it} + f_3 INFR_{it} + f_4 STRDEDGDP_{it}$$
$$+ f_5 S\&PINDEX_{it} + (\mu'_i + \tau_{it})$$
(12)

The selection of the better suited model from the two is done following the Hausman test as presented by [19], which appears thus:

$$H_{STAT} = (\beta^{FE} - \beta^{RE})'[Var(\beta^{FE}) - Var(\beta^{RE}]^{-1}(\beta^{FE} - \beta^{RE}) \sim \chi^{2(k)}$$
(13)

The Hausman test represents a distance measure between a fixed effect and random effect model with an $H_o$ that the random effects are better, efficient and consistent and an $H_1$ that the fixed effects (LSDV) are better, more efficient and consistent.

Fourth, we estimate Eqs 2 and 3 in a country-specific manner using the autoregressive distributed lag model by [20]. The procedure estimates simultaneously the long- and the short-run elasticity of the regressand to the modelled regressors. In addition, the [20] procedure employs the bounds testing approach to cointegration which is a generalized Dickey-Fuller type regression and tests the significance of the lagged level of the variables in a conditionally unrestricted error correction model (ECM).

The general equation of ARDL, (p,q) bounds test for co-integration is as follows:

$$\Delta CPSGDP_t = \alpha_1 + D_{jt} + \sum_{i=1}^{p} \delta_i \Delta CPSGDP_{t-i} + \sum_{j=0}^{q} \tau_j \Delta MCAPGDP_{t-j}$$
$$+ \sum_{i=1}^{p} \vartheta_k S\&PGLOBAL + \sum_{j=0}^{q} \theta_j \Delta STRDEDGDP_{t-j} + \sum_{j=0}^{q} \beta_j \Delta INFR_{t-j}$$
$$+ \omega_p CPSGDP_{t-1} + \omega_q MCAPGDP_{t-1} + \gamma_p S\&PGLOBAL_{t-1}$$
$$+ \varphi_i STRDEDGDP_{t-1} + \delta_k MCINFR_{t-1} + \xi_1$$
(14)

$$\Delta MCAPGDP_t = \alpha_1 + D_{jt} + \sum_{i=1}^{p} \delta_i \Delta MCAPGDP_{t-i} + \sum_{j=0}^{q} \tau_j \Delta CPSGDP_{t-j}$$

$$+ \sum_{i=1}^{p} \vartheta_k S\&PGLOBAL + \sum_{j=0}^{q} \theta_j \Delta STRDEDGDP_{t-j} + \sum_{j=0}^{q} \beta_j \Delta INFR_{t-j} \qquad (15)$$

$$+ \omega'_\rho MCAPGDP_{t-1} + \omega_q MCPSGDP_{t-1} + \gamma_p S\&PGLOBAL_{t-1}$$

$$+ \varphi_i STRDEDGDP_{t-1} + \delta_k MCINFR_{t-1} + \tau_1$$

The coefficients $\delta_i$, $\tau_j$, $\vartheta_k$, $\theta_l$ and $\beta_j$ are the short-run coefficients and $\omega_p$, $\omega_q$, $\gamma_p$, $\varphi_i$ and $\delta_k$ are the long-run parameters. Specifications (14) and (15) are due to Pesaran, Shin and Smith (2001) where short run effects are embodied in the estimates of coefficients attached to first-differenced variables. The long-run effects are, for example, judged by the estimates of $\omega_q$, $\gamma_p$, $\varphi_i$ and $\delta_k$, and are each divided by $-\omega_p$. To test the presence of co-integration, [20] have proposed the F-test. The decision is based on two critical bounds; the upper and the lower one. When the F-statistic is greater than the upper bound, the null hypothesis of the absence of a levels relationship is rejected. This means that there is evidence of cointegration.

In the second step, we followed the approach by [50, 51] and estimated the error-correction models and tested the statistical appropriateness of each model. This is supported by the relevant diagnostic tests to ensure, in particular, that the estimated models fulfil such major conditions as serial non-correlation, homoscedasticity and structural stability of the model.

## 4. Presentation and discussion of results

### 4.1 Results

Table 2 presents the results of the descriptive statistics, with emphasis on aggregative averages like the mean and measures of dispersion like the normal and relative standard deviations in both panel and country-specific dimensions.

We found that the average of the key bank development indicator (CPSGDP) for almost all the countries is less than the mean of the panel. This is with the exception of Brazil and China, whose average bank development indicators stand at 91.50 and 153.30, respectively, and are all greater than the panel mean of 88.80. The key indicators of spread and volatility for bank development, which are standard deviation and coefficient of variation, were found to be less in the countries than the panel, which stands at 39.22 and 0.44, respectively. The average MCAPGDP in the panel is also found to be greater than all the country-specific averages except in South Africa, with the highest mean MCAPGDP of 236.18. The dispersion following the coefficient of variation and standard deviation shows that all the countries are less volatile than the panel with a coefficient of variation and standard deviation of 4.93 and 41.79, respectively. The aggregative properties as well as the dispersion of the panel and country-specific interaction with the global equity index and inflation are also shown as part of the descriptive statistics summary.

Second, we present a summary of the panel unit root test and specific time series tests for all the countries in Table 3.

All the variables in panel form were found to be integrated of order zero I(0). In other words, they were all found to be stationary at levels at the 0.05 level of significance, which justifies the use of the conventional panel techniques. The country-specific unit root tests following the breakpoint consistent approach presents slightly mixed results. While all the variables in the countries showed stationarity at levels, the monetization ratio for South Africa (M2GDP)

**Table 2. Summary of panel and country specific descriptive statistics.**

| AVERAGES | CPSGDP | INFR | M2GDP | MCAPGDP | SPGLOBALEQUITY |
|---|---|---|---|---|---|
| **BRAZIL** | | | | | |
| MEAN | 91.50 | 6.45 | 73.0 | 47.95 | 13.18 |
| MAXIMUM | 113.71 | 14.71 | 96.14 | 80.22 | 125.11 |
| MINIMUM | 70.68 | 3.45 | 46.49 | 24.95 | -57.18 |
| STD DEVIATION | 14.54 | 2.74 | 14.09 | 14.84 | 49.85 |
| COEF OF VARIATION | 0.16 | 0.42 | 0.19 | 0.31 | 3.78 |
| **RUSSIA** | | | | | |
| MEAN | 48.56 | 7.47 | 54.16 | 36.60 | -0.20 |
| MAXIMUM | 58.84 | 15.53 | 61.82 | 62.38 | 106.63 |
| MINIMUM | 34.12 | 2.88 | 47.25 | 18.73 | -48.98 |
| STD DEVIATION | 8.91 | 3.39 | 5.16 | 12.35 | 34.59 |
| COEF OF VARIATION | 0.18 | 0.45 | 0.09 | 0.34 | 172.95 |
| **INDIA** | | | | | |
| MEAN | 70.76 | 7.61 | 75.14 | 75.58 | 16.61 |
| MAXIMUM | 88.43 | 11.99 | 79.07 | 149.51 | 94.14 |
| MINIMUM | 56.66 | 2.49 | 63.18 | 45.93 | -64.14 |
| STD DEVIATION | 7.41 | 2.89 | 5.09 | 27.31 | 53.98 |
| COEF OF VARIATION | 0.10 | 0.38 | 0.07 | 0.36 | 3.25 |
| **CHINA** | | | | | |
| MEAN | 153.38 | 2.67 | 172.80 | 50.59 | 6.81 |
| MAXIMUM | 218.31 | 5.92 | 202.96 | 126.15 | 80.72 |
| MINIMUM | 118.84 | -0.73 | 148.84 | 17.58 | -52.70 |
| STD DEVIATION | 32.19 | 1.92 | 19.68 | 22.66 | 39.32 |
| COEF OF VARIATION | 0.21 | 0.72 | 0.11 | 0.45 | 5.77 |
| **SOUTH AFRICA** | | | | | |
| MEAN | 76.22 | 5.74 | 69.75 | 236.18 | 6.00 |
| MAXIMUM | 84.42 | 10.05 | 80.80 | 352.16 | 53.74 |
| MINIMUM | 64.85 | -0.69 | 52.49 | 122.33 | -41.71 |
| STD DEVIATION | 5.22 | 1.84 | 9.08 | 62.11 | 25.74 |
| COEF OF VARIATION | 0.07 | 0.32 | 0.13 | 0.26 | 4.29 |
| **PANEL** | | | | | |
| MEAN | 88.08 | 5.99 | 88.97 | 89.38 | 8.48 |
| MAXIMUM | 218.31 | 15.53 | 202.96 | 352.16 | 125.11 |
| MINIMUM | 34.12 | -0.73 | 46.49 | 17.58 | -64.14 |
| STD DEVIATION | 39.22 | 3.15 | 44.31 | 81.58 | 41.79 |
| COEF OF VARIATION | 0.44 | 0.52 | 0.50 | 0.91 | 4.93 |

following the additive outlier showed stationarity at first difference, I(1). This mixed stationarity property justified the use of the autoregressive distributed lag regression technique given that it tolerates a combination of I(0) and I(1) excluding I(2).

Thirdly, We examined the likelihood of multicollinearity using several measures namely: correlation matrix for the panel data; condition number, variance inflation factor and tolerance ratio for the time series.

Table 4 below contains the results of the tests of the variables under study. The result shows that the variables share varied correlational relationship but of note is the fact that the correlational matrix refutes the possibility of multicollinearity among the studied variables. In other words, there is no high correlational coefficient as observed, which can lead to the conclusion

**Table 3. Summary of panel and country specific (time series) unit root tests.**

| VARIABLES | PANEL UNIT ROOT (LLC) | BRAZIL | | RUSSIA | | INDIA | | CHINA | | SOUTH AFRICA | |
|---|---|---|---|---|---|---|---|---|---|---|---|
| | | IO | AO | IO | AO | IO | AO | IO | AO | IO | AO |
| CPSGDP | t = -6.13 | t = -5.89 | t = -6.93 | t = -5.21 | t = -5.68 | t = -6.00 | t = -5.56 | t = -6.45 | t = -7.14 | t = -7.57 | t = -6.47 |
| | P<0.05 | P<0.05 | P<0.05 | P<0.05 | P<0.05 | P<0.05 | P<0.05 | P<0.05 | P<0.05 | P<0.05 | P<0.05 |
| | I(0) | I(0) | I(0) | I(0) | I(0) | I(0) | I(0) | I(0) | I(0) | I(0) | I(0) |
| INFR | t = -4.82 | t = -6.37 | t = -6.60 | t = -7.27 | t = -7.38 | t = -6.72 | t = -7.40 | t = -6.61 | t = -6.74 | t = -7.89 | t = -8.33 |
| | P<0.05 | P<0.05 | P<0.05 | P<0.05 | P<0.05 | P<0.05 | P<0.05 | P<0.05 | P<0.05 | P<0.05 | P<0.05 |
| | I(0) | I(0) | I(0) | I(0) | I(0) | I(0) | I(0) | I(0) | I(0) | I(0) | I(0) |
| M2GDP | t = -6.18 | t = -6.91 | t = -7.06 | t = -5.32 | t = -5.79 | t = -7.53 | t = -6.41 | t = -6.12 | t = -6.65 | t = -6.38 | t = -10.78 |
| | P<0.05 | P<0.05 | P<0.05 | P<0.05 | P<0.05 | P<0.05 | P<0.05 | P<0.05 | P<0.05 | P<0.05 | P<0.05 |
| | I(0) | I(0) | I(0) | I(0) | I(0) | I(0) | I(0) | I(0) | I(0) | I(0) | I(1) |
| MCAPGDP | t = -6.39 | t = -6.72 | t = -7.58 | t = -6.20 | t = -6.48 | t = -5.50 | t = -5.52 | t = -5.46 | t = -6.06 | t = -5.58 | t = -5.99 |
| | P<0.05 | P<0.05 | P<0.05 | P<0.05 | P<0.05 | P<0.05 | P<0.05 | P<0.05 | P<0.05 | P<0.05 | P<0.05 |
| | I(0) | I(0) | I(0) | I(0) | I(0) | I(0) | I(0) | I(0) | I(0) | I(0) | I(0) |
| SPGLOBALEQUITY | t = -5.97 | t = -7.16 | t = -5.63 | t = -6.35 | t = -7.64 | t = -7.88 | t = -9.08 | t = -7.00 | t = -7.30 | t = -7.06 | t = -7.63 |
| | P<0.05 | P<0.05 | P<0.05 | P<0.05 | P<0.05 | P<0.05 | P<0.05 | P<0.05 | P<0.05 | P<0.05 | P<0.05 |
| | I(0) | I(0) | I(0) | I(0) | I(0) | I(0) | I(0) | I(0) | I(0) | I(0) | I(0) |

of perfect collinearity or near singularity of the series under study. Also, all the condition numbers are below 30 and the variance inflation factors are not more than 5. Therefore the multicollinearity diagnostics show that it is within acceptable limits. Given that the variance-covariance matrix can be inverted in this study, multicollinearity appears to have a tolerable effect and has even further been contained by transformation of the series, (See [52]).

Third, we present the result of the conventional panel estimators following the fixed effect protocol as well as the random effect approach. The result is shown in Table 5.

To select the appropriate estimator that will be comparatively discussed with the country-specific ARDL estimates, the result of the Hausman test was used. In both models one and two as shown above, the Hausman test result was found to be highly significant, causing a rejection of the null hypothesis of efficiency of random effect estimators in favour of the efficiency of the fixed effect estimators. The panel results shown as part of Table 5 for the purposes of panel and country-specific comparative discussions follow the fixed effect form as supported by the Hausman tests.

Fourth, the verified panel estimators for the two models as well as the short-run and long-run elasticities in the country-specific estimations are shown below. The bounds test estimates, the panel cointegration results and the error-correction representation in addition to the diagnostic tests conducted are reported in Table 6.

To show the direction of causality among the studied variables, the Vector Autoregressive Causality and block exogeneity test is reported in Table 7 below:

The result of the VAR block exogeneity test as shown in Table 7, confirms with overwhelming empirical evidence the bidirectional interaction between bank development and stock market performance in the studied BRICS economies.

## 4.2 Bank development and stock market development

Following the panel estimators, bank development (CPSGDP) was found to be a positively significant function of stock market performance (MCAPGDP). Every unit change in stock market capitalization produced 7.5% increase in bank development. This is the same for all the countries following the ARDL estimates except for Russia, China and South Africa. The

**Table 4. Country specific and panel correlational/multicollinearity diagnostics.**

**BRAZIL**

| Variables | Model 1 | | | Model 2 | | |
|---|---|---|---|---|---|---|
| | Condition Numbers | T.R | VIF | Conditional Numbers | T.R | VIF |
| LINFR | 3.298 | .345 | 2.903 | 3.512 | .321 | 3.117 |
| LM2GDP | 9.223 | .118 | 8.441 | 9.836 | .034 | 29.240 |
| LSPGLOBALEQUITY | 19.171 | .247 | 4.048 | 20.606 | .242 | 4.129 |

**RUSSIA**

| Variables | Model 1 | | | Model 2 | | |
|---|---|---|---|---|---|---|
| | Condition Numbers | T.R | VIF | T.R | VIF | T.R |
| LINFR | 3.805 | .329 | 3.040 | 4.052 | .262 | 3.816 |
| LM2GDP | 13.421 | .305 | 3.274 | 14.215 | .045 | 22.221 |
| LSPGLOBALEQUITY | 15.608 | .843 | 1.186 | 14.402 | .793 | 1.261 |

**INDIA**

| Variables | Model 1 | | | Model 2 | | |
|---|---|---|---|---|---|---|
| | Condition Numbers | T.R | VIF | T.R | VIF | T.R |
| LINFR | 6.058 | .363 | 2.753 | 6.469 | .326 | 3.064 |
| LM2GDP | 14.037 | .368 | 2.718 | 14.689 | .093 | 10.755 |
| LSPGLOBALEQUITY | 22.450 | .608 | 1.644 | 23.807 | .600 | 1.666 |

**CHINA**

| Variables | Model 1 | | | Model 2 | | |
|---|---|---|---|---|---|---|
| | Condition Numbers | T.R | VIF | T.R | VIF | T.R |
| LINFR | 5.616 | .404 | 2.473 | 5.781 | .404 | 2.477 |
| LM2GDP | 6.548 | .308 | 3.247 | 6.977 | .103 | 9.691 |
| LSPGLOBALEQUITY | 11.486 | .707 | 1.414 | 12.311 | .653 | 1.530 |

**SOUTH AFRICA**

| Variables | Model 1 | | | Model 2 | | |
|---|---|---|---|---|---|---|
| | Condition Numbers | T.R | VIF | T.R | VIF | T.R |
| LINFR | 6.635 | .839 | 1.192 | 7.074 | .816 | 1.225 |
| LM2GDP | 16.301 | .323 | 3.100 | 17.455 | .120 | 8.366 |
| LSPGLOBALEQUITY | 20.096 | .783 | 1.277 | 21.134 | .752 | 1.329 |

**PANEL CORRELATION MATRIX**

| | CPSGDP | INFR | M2GDP | MCAPGDP | SPGLOBALEQUITY |
|---|---|---|---|---|---|
| CPSGDP | 1.000 | -0.531 | 0.938 | -0.117 | -0.050 |
| INFR | -0.531 | 1.000 | -0.535 | -0.031 | -0.062 |
| M2GDP | 0.938 | -0.535 | 1.000 | -0.158 | -0.027 |
| MCAPGDP | -0.117 | -0.031 | -0.158 | 1.000 | 0.145 |
| SPGLOBALEQUITY | -0.050 | -0.062 | -0.027 | 0.145 | 1.000 |

Where T.R = Tolerance Ratio, VIF = Variance Inflation Factor.

elasticity of bank development to stock market capitalization stands at 21.3% for Brazil and 9% for India, with Russia, China and South Africa showing negative and significant elasticities of 22%, 79% and 9%, respectively. There is a deviation between the development of China's stock market and the development of banks. In other words, banks are developing rapidly, but the stock market is not performing well. The internal mechanism lies in the state's control of banks, which leads to inefficient banks. However, due to the low quality of listed companies and the imperfect accounting system, especially the fraud of financial statements, the stock market has seriously affected the healthy development of the Chinese stock market".

**Table 5. Summary of the fixed effect and random effect estimators and Hausman tests.**

| Estimators | MODEL 1 | | | | | | MODEL 2 | | | | | |
|---|---|---|---|---|---|---|---|---|---|---|---|---|
| | Fixed effect | | | Random Effect | | | Fixed effect | | | Random Effect | | |
| | Coeff. | Std. Error | t-stat | Coeff. | Std. Error | t-stat | Coeff. | Std. Error | t-stat | Coeff. | Std. Error | t-stat |
| MCAPGDP | -0.075 | 0.026 | 2.84 (.005) | .028 | .017 | 1.69 (0.091) | NA | NA | NA | NA | NA | NA |
| CPSGDP | NA | NA | NA | NA | NA | NA | -0.753 | 0.264 | 2.84 (0.005) | .713 | .422 | 1.69 (0.091) |
| INFR | -0.22 | .266 | 0.83 (.406) | -.432 | .430 | 1.01 (0.314) | 0.297 | .845 | 0.35 (0.726) | -3.562 | 2.146 | 1.66 (0.097) |
| M2DGP | 1.30 | .067 | 19.45 (0.00) | .858 | .042 | 20.48 (0.000) | 1.385 | .396 | 3.49 (0.001) | -1.790 | .394 | 4.54 (0.000) |
| SPGLOBALEQUITY | 0.001 | 0.19 | 0.07 (0.94) | -.029 | .027 | 1.05 (0.292) | .290 | .053 | 5.44 (0.000) | .181 | .137 | 1.33 (0.185) |
| | *R²*: | | | *R²*: | | | *R²*: | | | *R²*: | | |
| | *Within = 0.77* | | | *Within = 0.74* | | | *Within = 0.41* | | | *Within = 0.21* | | |
| | *Between = 0.91* | | | *Between = 0.93* | | | *Between = 0.23* | | | *Between = 0.72* | | |
| | *Overall = 0.86* | | | *Overall = 0.88* | | | *Overall = 0.21* | | | *Overall = 0.33* | | |
| | *F-Stat = 91.31 (0.000)* | | | *Wald = 1073.65(0.000)* | | | *F-Stat = 18.99 (0.000)* | | | *Wald = 68.69(0.000)* | | |
| *Hausman Stat 64.5 (0.0000)* | | | | | | | *Hausman Stat 21.92 (0.0005)* | | | | | |
| *Choice Model: Fixed Effects* | | | | | | | *Choice Model: Fixed Effects* | | | | | |

On the other hand, there is a reverse positive causation by market capitalization on bank development, as shown in model 2 of the panel estimators, with a positive 75.3% degree of elasticity. The country-specific coefficients of elasticity also showed that market capitalization is a positive and significant function of bank development at the 0.05 level of significance, for Brazil, Russia, India and China, with the only exception being South Africa with a negatively significant elasticity coefficient.

Using both the Panel and ARDL estimators, we found that stock market performance drives bank development with a reverse causation and appropriate feedback.

## 4.3 Bank development and stock market development–cointegration and error- correction representation

We checked for cointegration using panel cointegration for the panel framework and the bounds test for the ARDL estimators. In all the models in both panel and country-specific framework as reported in the lower part of Table 4, we found evidence in favour of a long-run cointegrating relationship between bank development and market performance on one hand and market performance and bank development on the other hand. While the panel cointegration test for rejoinder models 1 and 2 were found to be significant, the ARDL bounds test for all the countries as reported are all greater than the upper band at the 0.05 level of significance.

We turned to a significant part of the results as shown in Table 4, that is, the short-run dynamics, which is the coefficient of the lagged error-correction term. A significant negative coefficient is expected if there is an adjustment towards the long-run equilibrium from short-run shocks. The absolute value of such a coefficient demonstrates how quickly or slowly variables return to the long-run equilibrium after a short-run shock, with particular attention given to the time span as indicated by the data frequency. For bank development and market performance models respectively, all the error-correction terms are negatively significant, indicating that they are rightly signed. There are no suspicions of oscillatory explosion because all of them fall within predictable limits (<1 or unity). We found that Brazil has the highest speed of adjustment at 92% for bank development and 81% for market performance. This is followed by China, with 89% speed of adjustment for bank development and 87% for market performance. This implies that in both China and Brazil, deviations from equilibrium in both bank development and market performance are restored in slightly above 1 year. Russia and India

**Table 6. Summary of short run and long run panel and country specific estimates.**

*MODEL 1: CPSGDP = f(M2GDP,INFR,MCAPGDP,SPLGLOBALEQUITY*
*MODEL 2: MCAPGDP = f(M2GDP,INFR,CPSGDP,SPLGLOBALEQUITY*

| VARIABLES | PANEL USING THE EFFICIENT ESTIMATES | | BRAZIL | | RUSSIA | | INDIA | | CHINA | | SOUTH AFRICA | |
|---|---|---|---|---|---|---|---|---|---|---|---|---|
| | MODEL 1 | MODEL 2 | MODEL 1 | MODEL 2 | MODEL 1 | MODEL 2 | MODEL 1 | MODEL 2 | MODEL 1 | MODEL 2 | MODEL 1 | MODEL 2 |
| CPSGDP | NA | $\alpha$ = -.753; t = 2.84 p<0.005 | NA | $\alpha$ = 1.00; t = 2.24 p<0.05 | NA | $\alpha$ = -1.26; t = 2.24 p<0.05 | NA | $\alpha$ = 3.05; t = 3.06 p<0.05 | NA | $\alpha$ = 0.53; t = 4.15 p<0.05 | NA | $\alpha$ = -6.41; t = 5.67 p<0.05 |
| M2GDP | $\alpha$ = 1.30; t = 19.4 p<0.05 | $\alpha$ = -1.39; t = 3.49 p<0.05 | $\alpha$ = 1.11; t = 19.04 p<0.05 | $\alpha$ = -1.33; t = 2.78 p<0.05 | $\alpha$ = 1.53; t = 10.53 p<0.05 | $\alpha$ = 2.59; t = 3.24 p<0.05 | $\alpha$ = 1.04; t = 8.89 p<0.05 | $\alpha$ = -0.93; t = 0.61 p>0.05 | $\alpha$ = 1.71; t = 5.46 p<0.05 | $\alpha$ = -0.70; t = 2.76 p<0.05 | $\alpha$ = 0.69; t = 11.18 p<0.05 | $\alpha$ = 4.63; t = 6.31 p<0.05 |
| MCAPGDP | $\alpha$ = -.075; t = 2.84 p<0.05 | NA | $\alpha$ = 0.213; t = 3.10 p<0.05 | NA | $\alpha$ = -0.22; t = 1.94 p<0.1 | NA | $\alpha$ = 0.09; t = 2.46 p<0.05 | NA | $\alpha$ = -0.79; t = 2.67 p<0.05 | NA | $\alpha$ = -0.09; t = 4.97 p<0.05 | NA |
| INFR | $\alpha$ = -.222; t = 0.83 p>0.05 | $\alpha$ = .297; t = 0.35 p>0.05 | $\alpha$ = 0.419; t = 1.31 p>0.05 | $\alpha$ = -1.67; t = 2.23 p<0.05 | $\alpha$ = -0.74; t = 3.31 p<0.05 | $\alpha$ = -2.08; t = 6.06 p<0.05 | $\alpha$ = 0.02; t = 0.07 p>0.05 | $\alpha$ = -1.25; t = 0.83 p>0.05 | $\alpha$ = 1.03; t = 0.24 p>0.05 | $\alpha$ = 9.84; t = 5.11 p<0.05 | $\alpha$ = -0.06; t = 0.18 p>0.05 | $\alpha$ = -2.78; t = 1.09 p<0.05 |
| SPLGLOBALEQUITY | $\alpha$ = .001; t = 0.07 p>0.05 | $\alpha$ = 0.29; t = 5.44 p<0.05 | $\alpha$ = -0.065; t = 3.53 p<0.05 | $\alpha$ = 0.21; t = 5.72 p<0.05 | $\alpha$ = 0.05; t = 2.29 p<0.05 | $\alpha$ = 0.20; t = 8.09 p<0.05 | $\alpha$ = -0.02; t = 1.35 p>0.05 | $\alpha$ = 0.31; t = 3.68 p<0.05 | $\alpha$ = 0.11; t = 0.51 p>0.05 | $\alpha$ = 0.40; t = 4.74 p<0.05 | $\alpha$ = 0.06; t = 2.91 p<0.05 | $\alpha$ = 0.74; t = 4.89 p<0.05 |
| $R^2$ | 0.86 | 0.21 | 0.97 | 0.86 | 0.99 | 0.98 | 0.87 | 0.88 | 0.91 | 0.97 | 0.94 | 0.96 |
| $\bar{R}^2$ | NA | NA | 0.96 | 0.81 | 0.98 | 0.95 | 0.81 | 0.78 | 0.83 | 0.90 | 0.87 | 0.94 |
| F-STAT | 91.31 | 18.99 | 64.3 | 86.7 | 78.69 | 35.92 | 15.07 | 8.26 | 11.47 | 14.51 | 43.19 | 9.95 |
| Bound Test (F-Stat) | NA | NA | 128.36; I(1) = 3.34 | 34.31; I(1) = 3.34 | 17.93; I(1) = 3.79 | 6.79; I(1) = 3.79 | 24.74; I(1) = 3.34 | 4.29; I(1) = 3.79 | 22.91; I(1) = 3.79 | 18.07; I(1) = 3.79 | 3.61; I(1) = 3.34 | 61.04; I(1) = 3.34 |
| Panel Cointegration | t = -7.56; p<0.05 | t = -5.09; p<0.05 | NA | NA | NA | NA | NA | NA | NA | NA | NA | NA |
| BG-LM TEST | NA | NA | 1.43; p>0.05 | 0.57; p>0.05 | 0.16; p>0.05 | 0.31; p>0.05 | 1.63; p>0.05 | 2.48; p>0.05 | 0.14; p>0.05 | 1.79; p>0.05 | 0.007; p>0.05 | 0,16; p>0.05 |
| HET-ARCH | NA | NA | 3.28; p>0.05 | 1.65; p>0.05 | 0.07; p>0.05 | 0.32; p>0.05 | 0.39; p>0.05 | 0.19; p>0.05 | 0.15; p>0.05 | 0.30; p>0.05 | 1.51; p>0.05 | 0.0004; p>0.05 |
| $ECM_{t-1}$ | NA | NA | $\alpha$ = -0.92; t = 30.87 p<0.05 | $\alpha$ = -0.81; t = 16.12 p<0.05 | $\alpha$ = -0.73; t = 12.70 p<0.05 | $\alpha$ = -0.61; t = 7.70 p<0.05 | $\alpha$ = -0.75; t = 13.77 p<0.05 | $\alpha$ = -0.60; t = 6.63 p<0.05 | $\alpha$ = -0.89; t = 13.66 p<0.05 | $\alpha$ = -0.87; t = 13.27 p<0.05 | $\alpha$ = -0.50; t = 5.54 p<0.05 | $\alpha$ = -0.39; t = 22.09 p<0.05 |

**Table 7. VAR causality/ block exogeneity Wald tests.**

| | Independent Variables | | |
|---|---|---|---|
| Variables/Countries | Levels | First-differences | VECM |
| | CPSGDP←MCAPGDP | CPSGDP←MCAPGDP | CPSGDP←MCAPGDP |
| | CPSGDP→MCAPGDP | CPSGDP→MCAPGDP | CPSGDP→MCAPGDP |
| **Brazil** | 92.75 [27.18] | 126.08 [110.16] | 49.82 [37.79] |
| **Russia** | 34.69 [33.09] | 11.00 [54.37] | 55.54 [49.24] |
| **India** | 49.69 [49.56] | 52.81 [22.99] | 72.01 [32.33] |
| **China** | 47.89 [75.49] | 19.05 [40.86] | 18.43 [37.24] |
| **South Africa** | 49.69 [49.45] | 26.66 [64.65] | 29.55 [71.65] |

Notes: Each chi-square test statistic is statistically significant at the 5% level. For example, in the case of the null hypothesis that there is no causality running from MCAPGDP to LCPSGDP using the level variables is shown as the chi-square value of 92.75, whereas, in the reverse causation LCPSGDP to MCAPGDP is given as 27.18 in a squared bracket.

have a very nearly similar adjustment profile as indicated by the error-correction representation. While the error-correction terms stand at 73% for bank development and 61% for market performance, respectively, for Russia, India has 75% for bank development and 60% for market performance. In both countries the speed of adjustment to full equilibrium from short-run shock stand at nearly 2 years. South Africa presents the lowest speed of adjustment for all the studied countries. The speed of adjustment for bank development is 50% and 39% for market performance. This implies that in South Africa deviations from equilibrium in both bank development and market performance are restored in 2 years for bank development and about 3 years for stock market performance. A further elaboration of reverse causation, we employ the data in Eq (1) and Eq (2), as well as the VAR method to perform block exogeneity tests and jointly estimate the two equation. These test statistics are shown in Table 5. It should be noted that we used an alternative method discussed in [53] and they yielded the same conclusions.

From the results, we gather that all the test values are statistically significant, hence confirming both direct and reverse causation.

## 4.4 Bank development, stock market development and global equity index

We evaluated the interaction among bank development, market performance and the global equity index. The panel estimates indicate that bank development is not a significant function of the S&P Global Equity Index. In the ARDL estimates for the bank development and S&P Global Equity Index for the studied countries, China and India were found to be insignificant, with Brazil showing bank development as a negatively significant function of the S&P Global Equity Index. For Russia and South Africa, we found that a unit change in the S&P Global Equity Index produced 5% and 6% positive changes in bank development, respectively.

On the other hand, stock market performance was found to be positively and significantly responsive to a unit change in the S&P Global Equity Index by 29%. In the country-specific ARDL estimates, we found that all the local stock markets in the BRICS countries are positive and significant functions of the S&P Global Equity Index, showing it to be a positively significant influencer of stock market performance. Brazil showed a positive coefficient of 21%, Russia 20%, India 31%, China 40% and South Africa 74%, all with the 0.05 level of significance.

## 4.5 Bank development, stock market performance and inflation nexus

Though we used inflation as a moderator in our estimation framework, we also discussed the empirical evidence provided in our analyses, with emphasis on the impact of inflation on bank development and stock market development in both panel and country-specific form. In both the first and second models in the panel estimation, we found that inflation exerted little significant impact on both stock market and bank development. For the country-specific estimators, inflation also had no significant impact on bank development in all the countries except for Russia, where it showed that every unit change in inflation produced a 0.74 point negative impact on bank development. In its relationship with stock market performance no significant relationship is found except in the cases of Brazil, Russia and China, where inflation proved to negatively and significantly affects stock market performance. The negative elasticity coefficient stands at 1.67 point for Brazil, 2.08 points for Russia and 9.84 points for China.

The diagnostic tests were the last issue we examined. These results are reported in the lower rung of Table 4. The adjusted $R^2$ following the ARDL estimates for the bank development model are 96%, 98%, 81%, 83% and 87%, respectively, for Brazil, Russia, India, China and South Africa. While the adjusted $R^2$ shows reasonable goodness of fit, the F-statistics for all the countries, evidently show that the model is statistically significant. Also, the adjusted $R^2$ following the ARDL estimates for the market performance model stands at 81%, 95%, 78%, 90% and 94% for Brazil,

Russia, India, China and South Africa, respectively. The adjusted $R^2$ signifies a good fit, and the F-statistics for all the countries, evidently show that the models are statistically significant. We carried out other diagnostic tests. To ensure that residuals are free of autocorrelation, we used the Breusch and Godfrey (BG) Lagrange multiplier (LM) tests for the autoregressive residual process. The absence or otherwise of heteroscedastic residuals was confirmed using the autoregressive conditional heteroscedasticity (ARCH) test. As all the results presented above suggest that there is enough evidence in support of the compliance of the model to the key underlying assumptions for which cause the results can be adjudged efficient and the estimators unbiased.

# 5. Conclusions, implications of the study and areas for further research

## 5.1 Conclusions

In this paper, an assortment of econometric techniques has been used to study the interaction among bank development, stock market performance and global equity index with a focus on the BRICS countries covering the period 1990 to 2018. This study made a combination of panel estimators with country specific/time series estimators in unveiling the degree and direction of the relationship among the studied variables.

Some key conclusions are drawn on the basis of the findings arising from this study.

First, we found a bidirectional causation between bank development (CPSGDP) and stock market performance (MCAPGDP) in BRICS countries. This empirical evidence is supported by both the panel estimators and the ARDL estimators as reported. It is also worthy to note that though all the results agreed in terms of magnitude, we found inter-country variation in the sense of direction of influence. For example, the elasticity of bank development to stock market performance stands at 21.3% for Brazil and 9% for India, with Russia, China (The result relating to China should be taken with caution following a suggestion by an anonymous reviewer on the pitfalls of using CPSGDP as bank development indicator for the Chinese economy. This follows the difficulty of the China's private sector in accessing credits and the relative smallness of the said credits) and South Africa showing negative and significant elasticities of 22%, 79% and 9%, respectively. On the other hand, there is a positive reverse causation in the case of market performance on bank development. The country-specific coefficients of elasticity also show that market performance is a positive and statistically significant function of bank development at the 0.05 level of significance. This underscores the fact that, though the BRICS countries are batched in an economic integration, it is somewhat preposterous to assume that they are on the same platform to allow for the same interactive effect amongst their economic and financial variables.

This outcome is consistent with our apriori expectation of a positive direct and reverse impact of stock market on bank development. This finding agrees with [6] who found coevolution in a study of the Nigerian financial system [29] who argued that the bank and stock market nexus is non-monotonic rather complimentary. Other authors such as [12, 31, 35, 54] conclude in agreement with our findings that no economies such as the BRICS countries run on a 'pure' model and that the strength in bank development should reflect on stock market development as well.

Second, we found cointegration using the panel cointegration framework and the bounds test for the ARDL estimator. This largely proves that a long-run relationship of both direct and reverse nature exists between bank development and stock market performance. It further supports the complementarity and coevolution hypothesis in the stock market and bank development nexus as previously espoused by [6, 33–35].

Third, for the bank development and market performance models, respectively, though all the error-correction terms are negatively significant, indicating that they are rightly signed, we found a varied speed of adjustment for all the BRICS countries. We found that Brazil has the

highest speed of adjustment, at 92% for bank development and 81% for market performance. This is followed by China with 89% speed of adjustment for bank development and 87% for market performance. This implies that, in both China and Brazil, deviations from equilibrium in both bank development and market performance are restored in slightly above 1 year. Russia and India have nearly similar adjustment profiles, as indicated by the error-correction representation. While the error-correction terms stand at 73% for bank development and 61% for market performance, respectively, for Russia, India has 75% for bank development and 60% for market performance. In both countries the speed of adjustment to full equilibrium from short-run shock stands at nearly 2 years. South Africa presents the lowest speed of adjustment for all the studied countries. The speed of adjustment for bank development is 50% and 39% for market performance. This implies that in South Africa, deviations from equilibrium in both bank development and market performance are restored in 2 years for bank development and about 3 years for stock market performance, respectively. This also underscores the likelihood of an individual country's economic nuances influencing the reaction and interactions of economic variables in the case of the BRICS countries. This makes it imperative for policy makers, social commentators and researchers alike, to look at these countries by striking a balance between common threshold or measures and individual characteristic.

Fourth, in our evaluation of the interaction among BRICS countries' bank development, market performance and the global equity index, we found that stock market performance responded more to the global equity index than the bank development indicator. The panel estimates indicate that bank development is not a significant function of the S&P Global Equity Index. The ARDL estimates for the bank development and S&P Global Equity Index interaction showed that in China and India were found to be insignificant, with Brazil showing a bank development as a negatively significant function of the S&P Equity Index. For Russia and South Africa, we found that a unit change in the S&P Equity Index produced 5% and 6% positive changes in bank development, respectively. On the other hand, stock market performance was found to be positive and significantly responsive to a unit change in S&P Global Equity Index by 29% in the panel. Also, we found that, in all BRICS countries, the stock market performance indicator proved to be a positive and significant function of the S&P equity index. Brazil showed a positive coefficient of 21%, Russia 20%, India 31%, China 40% and South Africa 74%. [7] examined the structure of dependence shared between the stock markets of BRICS using S&P 500 stock returns, the WTI crude price, the gold price, the U.S. policy uncertainty index and the VIX index, finding that the global stock and commodity markets exert a strong impact on the BRICS stock markets. Our findings in consistency with [8, 37–39]; even [40] provide evidence in favour of the globalized nature of the markets in the BRICS countries and their exposure to the trends and tides in the global stock market. More so, it goes to strengthen the argument of a stretch and spread of the BRICS countries in terms of international equity market and portfolio investment.

Lastly, though inflation was used as a moderator, our analyses of the results relative to inflation showed that inflation tends to exhibit moderate impact on stock market performance, with none on bank development. This is, however, not the case for all the BRICS countries as only stock market in Brazil, China and Russia showed sensitivity of varied forms to inflation. This is evidence in favour of the hedging effect as espoused by [55] and supported by [56–58] in the study of the stock market inflation nexus.

## 5.2 Policy implication and areas for further research

We conclude that the evidence arising from this study brings a new insight into the literature on stock market development, bank development and global equity interactions. Considering

the key positions of banks and the stock market as well as the global significance of the BRICS countries, this study will in no small measure propel policies that are designed to open the BRICS economies to further global financialisation. It is further expected that greater depth can be forged in the bank and stock market interaction to further accentuate the growth rate and development of the BRICS economies.

Our finding further suggests that policy aimed at stabilizing the banking sector have the tendency to exert positive impact on stock market development because of the complementarity discovery. This implies that policies that ignore this bidirectional interactions become too skewed to produce balanced effects on the entire financial system. Also, appropriate monetary policy should be conducted in the BRICS economies through a formal process of inflation targeting to mitigate the spatial and temporal effect of inflation on both the stock market and the banking system alike.

It is expected that this study will stir further research interest in firstly, validating previous conclusions drawn in this topical area of finance and economics by deploying more vigorous and recent statistical approaches. Also, it is expected that greater attention be given to some country characteristics in future studies that may focus on BRICS countries. This will further reduce the error of generalization and aggregation bias as is common with prior studies.

## Author Contributions

**Conceptualization:** Ebere Ume Kalu.

**Data curation:** Ebere Ume Kalu, Florence Ifeoma Onaga.

**Formal analysis:** Ebere Ume Kalu, Felix Chukwubuzo Alio.

**Funding acquisition:** Okoro E. U. Okoro, Florence Ifeoma Onaga, Felix Chukwubuzo Alio.

**Methodology:** Ebere Ume Kalu.

**Project administration:** Florence Ifeoma Onaga, Felix Chukwubuzo Alio.

**Resources:** Okoro E. U. Okoro.

**Software:** Ebere Ume Kalu.

**Supervision:** Augustine C. Arize.

**Validation:** Augustine C. Arize.

**Writing – review & editing:** Augustine C. Arize, Felix Chukwubuzo Alio.

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
