## [Decision Letter · Decision Letter 0]

18 May 2020

PONE-D-20-10606

A Cross-Country and Country-Specific Modeling of Stock Market Performance, Bank Development and Global Equity Index in Emerging Market Economies: A Case of BRICS Countries

PLOS ONE

Dear Dr Kalu,

Thank you for submitting your manuscript to PLOS ONE. After careful consideration, we feel that it has merit but does not fully meet PLOS ONE’s publication criteria as it currently stands. Therefore, we invite you to submit a revised version of the manuscript that addresses the points raised during the review process.

The manuscript requires further serious improvements regarding to a greater extent the research methodology and empirical findings.

We would appreciate receiving your revised manuscript by Jul 02 2020 11:59PM. To enhance the reproducibility of your results, we recommend that if applicable you deposit your laboratory protocols in protocols.io, where a protocol can be assigned its own identifier (DOI) such that it can be cited independently in the future. For instructions see: http://journals.plos.org/plosone/s/submission-guidelines#loc-laboratory-protocols

We look forward to receiving your revised manuscript.

Kind regards,

Stefan Cristian Gherghina, PhD. Habil.

Academic Editor

PLOS ONE

3. Please include a copy of Table 5 which you refer to in your text on page 23.

Reviewers' comments:

Reviewer's Responses to Questions

**Comments to the Author**

1. Is the manuscript technically sound, and do the data support the conclusions?

Reviewer #1: Partly

Reviewer #2: Partly

Reviewer #3: Yes

2. Has the statistical analysis been performed appropriately and rigorously? 

Reviewer #1: No

Reviewer #2: No

Reviewer #3: Yes

3. Have the authors made all data underlying the findings in their manuscript fully available?

Reviewer #1: Yes

Reviewer #2: No

Reviewer #3: Yes

4. Is the manuscript presented in an intelligible fashion and written in standard English?

Reviewer #1: No

Reviewer #2: Yes

Reviewer #3: Yes

5. Review Comments to the Author

Reviewer #1: 1. The research question is interesting, but the writing is sloppy and unintelligible, and contains many errors. I believe that an improvement on the writing by an editorial expert will significantly increase the quality of the paper.

2. The introductory section guides readers to understand the motivation, the importance and the contribution of the paper through a winding road of one thousand miles without a clear route.

3. The section of literature review looks more like a summary on papers cited. This section should systematically analyze, properly synthesize and critically evaluate the literature to deliver a clear picture of the state of knowledge on the subject, and figure out the gaps and to which you would like to contribute.

4. I fail to find out what LCPSGDP shown in Eq. (1) and other equations represents. Should it be CPSGDP?

5. As the authors said, the developments of the banking and the stock market could affect each other, so that a model that can address the reverse causation is essential. However, the authors did not do so, even though they present two equations of each of the two developments as the dependent variable in their paper.

The authors simply run regressions for each of the two equations, which is not the right way to address the reverse causation (endogeneity). The two equations should be simultaneously estimated in a system. Some proper techniques such as VAR model and simultaneous equation model can be used.

Reviewer #2: It is of great practical significance to study the interactive relationship between stock markets and banks in BRICS countries.

But I have some comments:

(1) There is a deviation between the development of China's stock market and the development of banks. In other words, banks are developing rapidly, but the stock market is not performing well. The internal mechanism lies in the state's control of banks, which leads to inefficient banks. However, due to the low quality of listed companies and the imperfect accounting system, especially the fraud of financial statements, the stock market has seriously affected the healthy development of the Chinese stock market. Therefore, this article should strengthen the analysis of the internal mechanism.

It is recommended to read the following paper.

Fang, J., Lau, C. K. M., Lu, Z., Tan, Y., & Zhang, H. (2019). Bank performance in China: A Perspective from Bank efficiency, risk-taking and market competition. Pacific-Basin Finance Journal, 56, 290-309.

(2) “Credit to the Private Sector as a ratio of Gross Domestic Product (CPSGDP) is used as a proxy for bank development.” I think this indicator is very inappropriate to measure the development of Bank of China. Because it is not easy for China's private sector to obtain loans and its proportion is relatively small, this will lead to doubts about the reliability of the results.

(3) The development of the bank and the stock market is not only highly related to economic development, but also has some deviations. But the lack of consideration of economic development indicators in the article is a major flaw.

Henry, P. B. (2000). Stock market liberalization, economic reform, and emerging market equity prices. The Journal of Finance, 55(2), 529-564.

Baele, L., De Jonghe, O., & Vander Vennet, R. (2007). Does the stock market value bank diversification?. Journal of Banking & Finance, 31(7), 1999-2023.

Reviewer #3: This paper investigates the relationship between stock market development, banking sector development and global equity index using a sample of data from BRICs over the period 1990-2018. I have the following comments to the authors:

1. the whole paper needs to be proofread to get rid of typos and grammar issues.

2. starting from the introduction, the authors argue that "In the light of the above, our enquiry is driven by three key considerations" however, the authors in detail talked about, one... second, third, fourth..... this is a contradiction.

3. it is recommended that that authors should discuss the significance of this study and contributions of the work in a separate and clearer manner. in other words, why this topic is important for BRIC and can the authors provide some detail evidence for each of these four countries to illustrate the importance of this topic and then clearly talked about the gap and their contribution.

4. in terms of the literature review, it is recommended that hypothesis development should be also established in this section in accordance to the main aims/objectives of this work. also some of the related works in related to the relationship between stock market development and banking sector development are missing from the literature review, I just listed a few below:

Tan, Y., and Floros, C. (2012a). Bank profitability and inflation: the case of China. Journal of Economic Studies, 39, 675-696.

Tan, Y., and Floros, C. (2012b). Bank profitability and GDP growth in China: a note. Journal of Chinese Economic and Business Studies, 10, 267-273.

5. related to the measurement of variable, the banking sector development is measured by the ratio of credit to the private sector to GDP in the current study, while this is different from Tan (2016) who use the ratio of banking sector assets to GDP, the authors are expected to provide critical discussion on this.

Tan, Y. (2016). The impacts of risk and competition on bank profitability in China. Journal of International Financial Markets, Institutions and Money, 40, 85-110.

6. correlation matrix should be provided to test the issue of multicollinearity.

7. I can not see any critical discussion about the results through providing linkage and comparison with existing empirical studies

8. it is recommended that the authors should talk about three issues in the conclusion in a clearer manner including the summary of the significance, contributions of the study; main results; as well as the policy implications and areas of future research.

6. PLOS authors have the option to publish the peer review history of their article (what does this mean?). If published, this will include your full peer review and any attached files.

Reviewer #1: No

Reviewer #2: None

Reviewer #3: No

---

## [Author Response · Author response to Decision Letter 0]

15 Jul 2020

Reviewer 1: 

Comment #1 

There is a deviation between the development of China's stock market and the development of banks. In other words, banks are developing rapidly, but the stock market is not performing well. The internal mechanism lies in the state's control of banks, which leads to inefficient banks. However, due to the low quality of listed companies and the imperfect accounting system, especially the fraud of financial statements, the stock market has seriously affected the healthy development of the Chinese stock market. Therefore, this article should strengthen the analysis of the internal mechanism.

It is recommended to read the following paper.

Response to Comment #1

We thank the reviewer for the above comment which we consider as a very helpful insight into an area of the stock market and bank development nexus in China that we overlooked. Useful additions have been made to the study by not just taking into account the comment but also reading the recommended papers out of which some additions were made to our study, see pages 9 and 21. 

Comment #2 

 “Credit to the Private Sector as a ratio of Gross Domestic Product (CPSGDP) is used as a proxy for bank development.” I think this indicator is very inappropriate to measure the development of Bank of China. Because it is not easy for China's private sector to obtain loans and its proportion is relatively small, this will lead to doubts about the reliability of the results.

Response to Comment #2

We followed prior authors using CPSGDP as a proxy for bank development. These include Rajan and Zingales in American Economic Review (1996) Levine in Journal of Economic Literature (1997), even more recent studies like Gaies in Energy Policy (2019), Maruta in Journal of Macroeconomics (2019), including three policy/working papers by the IMF on: (i) Remittances, Financial Development and Growth by Paola Giuliano and Marta Ruiz-Arranz (2006); (ii) Financial Development and Economic Growth by Jose De Gregorio and Pablo E. Guidotti (1995) and, iii. Introducing a New Broad-based Index of Financial Development by Katsiaryna Svirydzenka (2016). Undoubtedly, we appreciate the help of the reviewer in pointing out this nuance of the Chinese economy that is of great interest and we noted it on page 10. Suffice it to say that this point should be kept in mind by the readers. The use of CPSGDP here is for comparative and cross-country study purposes. Future studies that are specifically focused on the Chinese economy would represent a good enough platform for exploring this line of measurement for bank development. 

Comment #3 

The development of the bank and the stock market is not only highly related to economic development, but also has some deviations. But the lack of consideration of economic development indicators in the article is a major flaw.

Response to Comment #3

There is undoubtedly a relationship among bank, stock market and economic development as pointed out by the reviewer. We are thankful for this keen observation. We see financial development as a part of economic development which is of interest to this study, hence we explored bank development and stock market development as disaggregated components of financial development. It is also notable that financial development is a subset of economic development. Moreover, in measuring bank development and stock market development, we scaled credit to the private sector and market capitalization by GDP which is more or less a correlate of economic development. Nevertheless, it needs to be kept in mind that since a unique linear combination of the used variables produced cointegration, it is plausible that an excluded nonstationary variable such as economic development has little or no long run effect. 

In addition, the used ARDL approach minimizes problems associated with omitted variables. 

Reviewer 2:

Comment #1. 

The research question is interesting, but the writing is sloppy and unintelligible, and contains many errors. I believe that an improvement on the writing by an editorial expert will significantly increase the quality of the paper.

Response to Comment #1. 

We gratefully acknowledge the constructive and helpful comments provided by this referee. We believe that sufficient care has been taken to address the referee’s comments. Most importantly, the paper has greater clarity because of the reviewer’s helpful comments. For this, the authors wish to thank the referee.

Comment #2. 

The introductory section guides readers to understand the motivation, the importance and the contribution of the paper through a winding road of one thousand miles without a clear route.

Response to Comment #2. 

The paper also benefited from fruitful comments concerning the introduction. We have retouched it to make it more pointed and better directed. We are very grateful to the referee. 

Comment #3. 

The section of literature review looks more like a summary on papers cited. This section should systematically analyze, properly synthesize and critically evaluate the literature to deliver a clear picture of the state of knowledge on the subject, and figure out the gaps and to which you would like to contribute.

Response to Comment #3. 

The literature section has been rewritten and it now looks better than the original submission, thanks to the reviewers.

Comment #4.

4. I fail to find out what LCPSGDP shown in Eq. (1) and other equations represents. Should it be CPSGDP?

Response to Comment #4. 

The referee is correct and we have fixed that by correcting the interchangeable use of CPSGDP in its level and log forms has been corrected and a properly harmonized presentation made in the revised manuscript.

Comment #5.

As the authors said, the developments of the banking and the stock market could affect each other, so that a model that can address the reverse causation is essential. However, the authors did not do so, even though they present two equations of each of the two developments as the dependent variable in their paper.

The authors simply run regressions for each of the two equations, which is not the right way to address the reverse causation (endogeneity). The two equations should be simultaneously estimated in a system. Some proper techniques such as VAR model and simultaneous equation model can be used.

Response to Comment #5.

Following the referee’s comments, we have tested endogeneity using the VAR method and the result is in table 5 page 20. 

General Comment

The comments by the Editor were also addressed in the revised manuscript.

This study as revised is obviously better than its original form, thanks to the reviewers for their useful and very valid comments. The authors unreservedly express their gratitude to them for this good work and most of all to the editor for his uncommon ability in driving the paper through the publication process.

---

## [Decision Letter · Decision Letter 1]

3 Aug 2020

PONE-D-20-10606R1

A Cross-Country and Country-Specific Modeling of Stock Market Performance, Bank Development and Global Equity Index in Emerging Market Economies: A Case of BRICS Countries

PLOS ONE

Dear Dr. Kalu,

Thank you for submitting your manuscript to PLOS ONE. After careful consideration, we feel that it has merit but does not fully meet PLOS ONE’s publication criteria as it currently stands. Therefore, we invite you to submit a revised version of the manuscript that addresses the points raised during the review process.

The manuscript requires further amendments in as much as the entire reviewers’ comments and suggestions should be implemented. Actually, the recommendations formulated by Reviewer 3 were not considered.

We look forward to receiving your revised manuscript.

Kind regards,

Stefan Cristian Gherghina, PhD. Habil.

Academic Editor

PLOS ONE

Reviewers' comments:

Reviewer's Responses to Questions

**Comments to the Author**

1. If the authors have adequately addressed your comments raised in a previous round of review and you feel that this manuscript is now acceptable for publication, you may indicate that here to bypass the “Comments to the Author” section, enter your conflict of interest statement in the “Confidential to Editor” section, and submit your "Accept" recommendation.

Reviewer #1: All comments have been addressed

Reviewer #2: All comments have been addressed

Reviewer #3: (No Response)

2. Is the manuscript technically sound, and do the data support the conclusions?

Reviewer #1: Partly

Reviewer #2: Partly

Reviewer #3: (No Response)

3. Has the statistical analysis been performed appropriately and rigorously? 

Reviewer #1: (No Response)

Reviewer #2: Yes

Reviewer #3: (No Response)

4. Have the authors made all data underlying the findings in their manuscript fully available?

Reviewer #1: Yes

Reviewer #2: No

Reviewer #3: (No Response)

5. Is the manuscript presented in an intelligible fashion and written in standard English?

Reviewer #1: (No Response)

Reviewer #2: Yes

Reviewer #3: (No Response)

6. Review Comments to the Author

Reviewer #1: (No Response)

Reviewer #2: The font size and format of Table 1 and Table 2, 3 in the paper are not uniform. The author's revision basically meets the reviewers' requirements, although reviewers still have reservations about the selection of some indicators.

Reviewer #3: It seems that the author did not address any of my comments. therefore, I suggest a rejection. I was reviewer 3 in the last round, no effort has been shown to address my comments in this revision.

7. PLOS authors have the option to publish the peer review history of their article (what does this mean?). If published, this will include your full peer review and any attached files.

Reviewer #1: No

Reviewer #2: No

Reviewer #3: No

---

## [Author Response · Author response to Decision Letter 1]

18 Aug 2020

Reviewer 1: 

Comment #1 

There is a deviation between the development of China's stock market and the development of banks. In other words, banks are developing rapidly, but the stock market is not performing well. The internal mechanism lies in the state's control of banks, which leads to inefficient banks. However, due to the low quality of listed companies and the imperfect accounting system, especially the fraud of financial statements, the stock market has seriously affected the healthy development of the Chinese stock market. Therefore, this article should strengthen the analysis of the internal mechanism.

It is recommended to read the following paper.

Response to Comment #1

We thank the reviewer for the above comment which we consider as a very helpful insight into an area of the stock market and bank development nexus in China that we overlooked. Useful additions have been made to the study by not just taking into account the comment but also reading the recommended papers out of which some additions were made to our study, see pages 9 and 21. 

Comment #2 

 “Credit to the Private Sector as a ratio of Gross Domestic Product (CPSGDP) is used as a proxy for bank development.” I think this indicator is very inappropriate to measure the development of Bank of China. Because it is not easy for China's private sector to obtain loans and its proportion is relatively small, this will lead to doubts about the reliability of the results.

Response to Comment #2

We followed prior authors using CPSGDP as a proxy for bank development. These include Rajan and Zingales in American Economic Review (1996) Levine in Journal of Economic Literature (1997), even more recent studies like Gaies in Energy Policy (2019), Maruta in Journal of Macroeconomics (2019), including three policy/working papers by the IMF on: (i) Remittances, Financial Development and Growth by Paola Giuliano and Marta Ruiz-Arranz (2006); (ii) Financial Development and Economic Growth by Jose De Gregorio and Pablo E. Guidotti (1995) and, iii. Introducing a New Broad-based Index of Financial Development by Katsiaryna Svirydzenka (2016). Undoubtedly, we appreciate the help of the reviewer in pointing out this nuance of the Chinese economy that is of great interest and we noted it on page 10. Suffice it to say that this point should be kept in mind by the readers. The use of CPSGDP here is for comparative and cross-country study purposes. Future studies that are specifically focused on the Chinese economy would represent a good enough platform for exploring this line of measurement for bank development. 

Comment #3 

The development of the bank and the stock market is not only highly related to economic development, but also has some deviations. But the lack of consideration of economic development indicators in the article is a major flaw.

Response to Comment #3

There is undoubtedly a relationship among bank, stock market and economic development as pointed out by the reviewer. We are thankful for this keen observation. We see financial development as a part of economic development which is of interest to this study, hence we explored bank development and stock market development as disaggregated components of financial development. It is also notable that financial development is a subset of economic development. Moreover, in measuring bank development and stock market development, we scaled credit to the private sector and market capitalization by GDP which is more or less a correlate of economic development. Nevertheless, it needs to be kept in mind that since a unique linear combination of the used variables produced cointegration, it is plausible that an excluded nonstationary variable such as economic development has little or no long run effect. 

In addition, the used ARDL approach minimizes problems associated with omitted variables. 

Reviewer 2:

Comment #1. 

The research question is interesting, but the writing is sloppy and unintelligible, and contains many errors. I believe that an improvement on the writing by an editorial expert will significantly increase the quality of the paper.

Response to Comment #1. 

We gratefully acknowledge the constructive and helpful comments provided by this referee. We believe that sufficient care has been taken to address the referee’s comments. Most importantly, the paper has greater clarity because of the reviewer’s helpful comments. For this, the authors wish to thank the referee.

Comment #2. 

The introductory section guides readers to understand the motivation, the importance and the contribution of the paper through a winding road of one thousand miles without a clear route.

Response to Comment #2. 

The paper also benefited from fruitful comments concerning the introduction. We have retouched it to make it more pointed and better directed. We are very grateful to the referee. 

Comment #3. 

The section of literature review looks more like a summary on papers cited. This section should systematically analyze, properly synthesize and critically evaluate the literature to deliver a clear picture of the state of knowledge on the subject, and figure out the gaps and to which you would like to contribute.

Response to Comment #3. 

The literature section has been rewritten and it now looks better than the original submission, thanks to the reviewers.

Comment #4.

4. I fail to find out what LCPSGDP shown in Eq. (1) and other equations represents. Should it be CPSGDP?

Response to Comment #4. 

The referee is correct and we have fixed that by correcting the interchangeable use of CPSGDP in its level and log forms has been corrected and a properly harmonized presentation made in the revised manuscript.

Comment #5.

As the authors said, the developments of the banking and the stock market could affect each other, so that a model that can address the reverse causation is essential. However, the authors did not do so, even though they present two equations of each of the two developments as the dependent variable in their paper.

The authors simply run regressions for each of the two equations, which is not the right way to address the reverse causation (endogeneity). The two equations should be simultaneously estimated in a system. Some proper techniques such as VAR model and simultaneous equation model can be used.

Response to Comment #5.

Following the referee’s comments, we have tested endogeneity using the VAR method and the result is in table 5 page 20. 

General Comment

The comments by the Editor were also addressed in the revised manuscript.

This study as revised is obviously better than its original form, thanks to the reviewers for their useful and very valid comments. The authors unreservedly express their gratitude to them for this good work and most of all to the editor for his uncommon ability in driving the paper through the publication process. 

Reviewer #3: 

Introduction

We most sincerely apologise to reviewer #3 for the inadvertent omission of our response to his excellent sets of comments. Permit me to say that it was not deliberate but an innocent omission. Reviewer 3 is among the best in terms of detailed comments and helpful review of our study and does not in any way deserve the regrettable omission of our response to the helpful comments. We have responded as requested to the 8 comments raised by the reviewer and undoubtedly the paper is better than the original and revised (R1) submission; thanks to the help of Reviewer 3.

Comment #1. 

The whole paper needs to be proofread to get rid of typos and grammar issues.

Response to Comment #1.

As suggested by the referee, we employed the services of a professional editor who proofread the entire paper at a significant cost. We are grateful to the referee for this suggestion.

Comment #2.

Starting from the introduction, the authors argue that "In the light of the above, our enquiry is driven by three key considerations" however, the authors in detail talked about, one... second, third, fourth..... this is a contradiction.

Response to Comment #2.

The observation by the reviewer is quite true and the observed contradiction in numbering the “considerations” has been put in order. We are grateful to the reviewer for his helpfulness.

Comment #3.

It is recommended that the authors should discuss the significance of this study and contributions of the work in a separate and clearer manner. In other words, why this topic is important for BRIC and can the authors provide some detail evidence for each of these four countries to illustrate the importance of this topic and then clearly talked about the gap and their contribution.

Response to Comment #3.

Following the reviewer’s suggestion, page 3 addresses the relevance of this study for the BRICS countries see also table 1 with the highlighted nuances of the BRICS countries and the discussions on how the study will be of significance to the respective BRICS countries. Many thanks to the reviewer for this comment.

Comment #4.

In terms of the literature review, it is recommended that hypothesis development should be also established in this section in accordance to the main aims/objectives of this work. also some of the related works in related to the relationship between stock market development and banking sector development are missing from the literature review, I just listed a few below:

Tan, Y., and Floros, C. (2012a). Bank profitability and inflation: the case of China. Journal of Economic Studies, 39, 675-696.

Tan, Y., and Floros, C. (2012b). Bank profitability and GDP growth in China: a note. Journal of Chinese Economic and Business Studies, 10, 267-273.

Response to Comment #4.

The literature section has entirely been rewritten to capture the stream of argument in the bank and stock market development interactions. More studies including the two suggested by the reviewer have been added to the literature section and has been made more robust than it was originally. In addition, the main thesis of the study has also been captured with the gap that this study intends to fill (see the last two paragraphs in the literature section pg. 9). We also included the three key hypotheses of interest to this study. Thanks to the reviewer for this comment. 

Comment #5.

Related to the measurement of variable, the banking sector development is measured by the ratio of credit to the private sector to GDP in the current study, while this is different from Tan (2016) who use the ratio of banking sector assets to GDP, the authors are expected to provide critical discussion on this.

Tan, Y. (2016). The impacts of risk and competition on bank profitability in China. Journal of International Financial Markets, Institutions and Money, 40, 85-110.

Response to Comment #5.

A discussion on this has been provided to justify the use of CPSGDP as a proxy for bank development. We thank the reviewer for making us provide a justification for the parameter used in measuring bank development (See footnote 1, page 10).

Comment #6.

Correlation matrix should be provided to test the issue of multicollinearity.

Response to Comment #6.

In response to the recommendation of reviewer 3, the correlation matrix has been added as table 4 in the study. Given that the variance-covariance matrix can be inverted following our result, multicollinearity appears to have a tolerable effect and has further been contained by transformation of the series (See Berndt, 1996: pp 339) .

Comment #7.

I cannot see any critical discussion about the results through providing linkage and comparison with existing empirical studies.

Response to Comment #7.

The result section has been rewritten to link the findings with our aprior expectation as well as existing empirical studies. Thanks to this reviewer for this excellent comment.

Comment #8.

It is recommended that the authors should talk about three issues in the conclusion in a clearer manner including the summary of the significance, contributions of the study; main results; as well as the policy implications and areas of future research.

Response to Comment #8.

The conclusion segment of this study has been disaggregated to capture the salient issues raised by this reviewer. It is our firm belief that the section has greater clarity and direction, thanks to the insightful comment by the reviewer.

---

## [Decision Letter · Decision Letter 2]

26 Aug 2020

PONE-D-20-10606R2

A Cross-Country and Country-Specific Modeling of Stock Market Performance, Bank Development and Global Equity Index in Emerging Market Economies: A Case of BRICS Countries

PLOS ONE

Dear Dr. Kalu,

Thank you for submitting your manuscript to PLOS ONE. After careful consideration, we feel that it has merit but does not fully meet PLOS ONE’s publication criteria as it currently stands. Therefore, we invite you to submit a revised version of the manuscript that addresses the points raised during the review process.

The author(s) should further consider the recommendations formulated by the third reviewer.

We look forward to receiving your revised manuscript.

Kind regards,

Stefan Cristian Gherghina, PhD. Habil.

Academic Editor

PLOS ONE

Reviewers' comments:

Reviewer's Responses to Questions

**Comments to the Author**

1. If the authors have adequately addressed your comments raised in a previous round of review and you feel that this manuscript is now acceptable for publication, you may indicate that here to bypass the “Comments to the Author” section, enter your conflict of interest statement in the “Confidential to Editor” section, and submit your "Accept" recommendation.

Reviewer #1: (No Response)

Reviewer #2: All comments have been addressed

Reviewer #3: (No Response)

2. Is the manuscript technically sound, and do the data support the conclusions?

Reviewer #1: (No Response)

Reviewer #2: Partly

Reviewer #3: (No Response)

3. Has the statistical analysis been performed appropriately and rigorously? 

Reviewer #1: (No Response)

Reviewer #2: Yes

Reviewer #3: (No Response)

4. Have the authors made all data underlying the findings in their manuscript fully available?

Reviewer #1: (No Response)

Reviewer #2: No

Reviewer #3: (No Response)

5. Is the manuscript presented in an intelligible fashion and written in standard English?

Reviewer #1: (No Response)

Reviewer #2: Yes

Reviewer #3: (No Response)

6. Review Comments to the Author

Reviewer #1: (No Response)

Reviewer #2: I have no more commments.

Reviewer #3: I do not think the authors seriously considered my comments in the revision, for example, my comment 4 was not addressed satisfactorily, also I do not think the authors addressed comment 6 in an appropriate way. finally, there are lots of missing references compared to related in-text citations.

7. PLOS authors have the option to publish the peer review history of their article (what does this mean?). If published, this will include your full peer review and any attached files.

Reviewer #1: No

Reviewer #2: No

Reviewer #3: No

---

## [Author Response · Author response to Decision Letter 2]

7 Sep 2020

Reviewer #3:

Comment #1.

In terms of the literature review, it is recommended that hypothesis development should be also established in this section in accordance to the main aims/objectives of this work. The hypotheses below were integrated as part of the literature section:

Ho1: Bank development does not drive stock market development

Ho2: Stock Market development does drive bank development

Ho3: Global equity index has no significant impact on bank and stock market development.

The above hypotheses were relocated to the beginning part of the literature section making them provide definite direction for the overall development of the literature review section.

Comment #2.

Correlation matrix should be provided to test the issue of multicollinearity.

Response to Comment #2.

In response to the recommendation of reviewer 3, the correlation matrix has been added as table 4 in the study. Given that the variance-covariance matrix can be inverted following our result, multicollinearity appears to have a tolerable effect and has further been contained by transformation of the series (See Berndt, 1996: pp 339).

Additionally, we created the country specific multicollinearity diagnostics using three measures:

-condition numbers

-Variance Inflation Factors

-Tolerance Ratio

All these measures agree with the earlier established position by the panel correlation matrix which is presented at the lower part of table 4.

Comment #3. 

Missing references compared to related in-text citations

Response to Comment #3.

The reference section has been revisited and the missing references have been added. We thank Reviewer 3 for his/her keen observation.

---

## [Decision Letter · Decision Letter 3]

28 Sep 2020

A Cross-Country and Country-Specific Modeling of Stock Market Performance, Bank Development and Global Equity Index in Emerging Market Economies: A Case of BRICS Countries

PONE-D-20-10606R3

Dear Dr. Kalu,

We’re pleased to inform you that your manuscript has been judged scientifically suitable for publication and will be formally accepted for publication once it meets all outstanding technical requirements.

Kind regards,

Stefan Cristian Gherghina, PhD. Habil.

Academic Editor

PLOS ONE

Additional Editor Comments (optional):

Reviewers' comments:

Reviewer's Responses to Questions

**Comments to the Author**

1. If the authors have adequately addressed your comments raised in a previous round of review and you feel that this manuscript is now acceptable for publication, you may indicate that here to bypass the “Comments to the Author” section, enter your conflict of interest statement in the “Confidential to Editor” section, and submit your "Accept" recommendation.

Reviewer #1: All comments have been addressed

Reviewer #2: All comments have been addressed

Reviewer #3: (No Response)

2. Is the manuscript technically sound, and do the data support the conclusions?

Reviewer #1: Partly

Reviewer #2: Partly

Reviewer #3: (No Response)

3. Has the statistical analysis been performed appropriately and rigorously? 

Reviewer #1: Yes

Reviewer #2: Yes

Reviewer #3: (No Response)

4. Have the authors made all data underlying the findings in their manuscript fully available?

Reviewer #1: No

Reviewer #2: Yes

Reviewer #3: (No Response)

5. Is the manuscript presented in an intelligible fashion and written in standard English?

Reviewer #1: Yes

Reviewer #2: Yes

Reviewer #3: (No Response)

6. Review Comments to the Author

Reviewer #1: (No Response)

Reviewer #2: This study examines the interplay between bank development, stock market development and global stock indexes in the BRICS countries from 1990 to 2018. This has certain significance for the development and policy making of BRICS National Bank and the stock market. All requirements by reviwers have been met.

Reviewer #3: (No Response)

7. PLOS authors have the option to publish the peer review history of their article (what does this mean?). If published, this will include your full peer review and any attached files.

Reviewer #1: No

Reviewer #2: No

Reviewer #3: No

---

## [Editor Report · Acceptance letter]

29 Oct 2020

PONE-D-20-10606R3 

A Cross-Country and Country-Specific Modelling of Stock Market Performance, Bank Development and Global Equity Index in Emerging Market Economies: A Case of BRICS Countries 

Dear Dr. Kalu:

I'm pleased to inform you that your manuscript has been deemed suitable for publication in PLOS ONE. Congratulations! Your manuscript is now with our production department. 

Kind regards, 

on behalf of

Dr. Stefan Cristian Gherghina 

Academic Editor

PLOS ONE